# The Impact of Age on In-Hospital Mortality in Critically Ill COVID-19 Patients: A Retrospective and Multicenter Study

**DOI:** 10.3390/diagnostics12030666

**Published:** 2022-03-09

**Authors:** Pierrick Le Borgne, Quentin Dellenbach, Karine Alame, Marc Noizet, Yannick Gottwalles, Tahar Chouihed, Laure Abensur Vuillaume, Charles-Eric Lavoignet, Lise Bérard, Lise Molter, Stéphane Gennai, Sabrina Kepka, François Lefebvre, Pascal Bilbault

**Affiliations:** 1Emergency Department, Hôpitaux Universitaires de Strasbourg, 67000 Strasbourg, France; quentin.dellenbach@chru-strasbourg.fr (Q.D.); karine.alame@chru-strasbourg.fr (K.A.); sabrina.garnier-kepka@chru-strasbourg.fr (S.K.); pascal.bilbault@chru-strasbourg.fr (P.B.); 2INSERM (French National Institute of Health and Medical Research), UMR 1260, Regenerative NanoMedicine (RNM), Fédération de Médecine Translationnelle (FMTS), University of Strasbourg, 67000 Strasbourg, France; 3CREMS Network (Clinical Research in Emergency Medicine and Sepsis), 67201 Wolfisheim, France; l.abensurvuillaume@chr-metz-thionville.fr (L.A.V.); charles-eric.lavoignet@hnfc.fr (C.-E.L.); lise.berard@ch-haguenau.fr (L.B.); 4Emergency Department, Mulhouse Hospital, 68100 Mulhouse, France; marc.noizet@ghrmsa.fr; 5Emergency Department, Colmar Hospital, 68026 Colmar, France; yannick.gottwalles@ch-colmar.fr; 6Emergency Department, Nancy University Hospital, 54500 Nancy, France; t.chouihed@chru-nancy.fr; 7Centre d’Investigations Cliniques-1433, and INSERM U1116, F-CRIN INI-CRCT, Université de Lorraine, 54000 Nancy, France; 8Emergency Department, Regional Hospital of Metz-Thionville, 57000 Metz, France; 9Emergency Department, Nord Franche Comté Hospital, 90400 Trevenans, France; 10Emergency Department, Haguenau Hospital, 67500 Haguenau, France; 11Emergency Department, Verdun Hospital, 55107 Verdun, France; lise_molter@hotmail.com; 12Emergency Department, Reims University Hospital, 51092 Reims, France; sgennai@chu-reims.fr; 13Department of Public Health, University Hospital of Strasbourg, 67200 Strasbourg, France; francois.lefebvre@chru-strasbourg.fr

**Keywords:** COVID-19, ICU, age, elderly

## Abstract

Introduction: For the past two years, healthcare systems worldwide have been battling the ongoing COVID-19 pandemic. Several studies tried to find predictive factors of mortality in COVID-19 patients. We aimed to research age as a predictive factor associated with in-hospital mortality in severe and critical SARS-CoV-2 infection. Methods: Between 1 March and 20 April 2020, we conducted a multicenter and retrospective study on a cohort of severe COVID-19 patients who were all hospitalized in the Intensive Care Unit (ICU). We led our study in nine hospitals of northeast France, one of the pandemic’s epicenters in Europe. Results: The median age of our study population was 66 years (58–72 years). Mortality was 24.6% (CI 95%: 20.6–29%) in the ICU and 26.5% (CI 95%: 22.3–31%) in the hospital. Non-survivors were significantly older (69 versus 64 years, *p* < 0.001) than the survivors. Although a history of cardio-vascular diseases was more frequent in the non-survivor group (*p* = 0.015), other underlying conditions and prior level of autonomy did not differ between the two groups. On multivariable analysis, age appeared to be an interesting predictive factor of in-hospital mortality. Thus, age ranges of 65 to 74 years (OR = 2.962, CI 95%: 1.231–7.132, *p* = 0.015) were predictive of mortality, whereas the group of patients aged over 75 years was not (OR = 3.084, CI 95%: 0.952–9.992, *p* = 0.06). Similarly, all comorbidities except for immunodeficiency (OR = 4.207, CI 95%: 1.006–17.586, *p* = 0.049) were not predictive of mortality. Finally, survival follow-up was obtained for the study population. Conclusion: Age appears to be a relevant predictive factor of in-hospital mortality in cases of severe or critical SARS-CoV-2 infection.

## 1. Introduction

In December 2019, the emergence of a novel coronavirus, severe acute respiratory syndrome coronavirus 2 (SARS-CoV-2), spread worldwide and caused an ongoing and devastating pandemic [1]. As of the 1 September 2021, this emerging virus has infected more than 210 million people resulting in over 4 million deaths worldwide. The clinical spectrum of this viral infection is broad. Once infected, the patients are, for a majority, asymptomatic or paucisymptomatic, presenting influenza-like symptoms. A certain number of patients develop mild disease and may require hospitalization for viral hypoxemic pneumonia. A minority of patients present severe or critical disease with complications such as acute respiratory distress syndrome (ARDS), requiring ventilatory support in the Intensive Care Unit (ICU) [2,3].

Since the outbreak started, several studies tried to find predictive factors of mortality in COVID-19 patients. [4,5]. Although patients over 65 years old made up only 17% of the total population in the United States of America, they also made up 31% of COVID-19 cases, 45% of admissions to conventional hospital units, 53% of ICU admissions, and almost 80% of deaths during the first wave of the pandemic [6]. Compared to younger patients, the elderly (defined by the WHO as 65 years and older) seem to have a higher risk of death along with decreased level of autonomy. Several reasons might explain why elderly patients have been so frequently impacted by this emerging infection. First, the clinical presentation is frequently atypical; hence, the diagnosis is often delayed. Second, quarantine and preventive health measures (especially for the most dependent patients in nursing homes) are not always ideal. Lastly, the association of frequent comorbidities, a dysfunction of the immune response and a reduction in the expression of ACE2 could explain the high susceptibility to infection and the heavy toll paid by the elderly during this pandemic [7].

Across the world, hospital resources, especially those of ICUs, were overwhelmed by a massive load of COVID-19 patients requiring mechanical ventilation during the first wave of the outbreak (1st semester 2020). Physicians were faced, on one hand, with ICU admission demands rapidly hitting crisis level, and, on the other hand, with no alternative ventilatory strategy than invasive mechanical ventilation. As a solution, temporary ICUs and nightingale hospitals were created, yet critical care strains had consequences on the quality of care provided and patient mortality [8]. An analysis of data from the first wave, when no specific treatment or vaccine was available, could be a fundamental growth moment, improving our capacity to face emergencies in future health crisis settings [9]. Therefore, we conducted this study on a cohort of severe and critical COVID-19 patients who were all admitted to the ICU after presenting to the emergency department (ED) of nine northeastern French hospitals during the onset of the pandemic’s first wave (March–May 2020). We aimed to study age as a predictive factor associated with in-hospital mortality in severe or critical SARS-CoV-2 infection.

## 2. Methods

### 2.1. Study Population and Settings

This retrospective, multicenter study was conducted in nine EDs in northeast France: three university hospitals (CHRU of Strasbourg, CHRU of Nancy and CHU of Reims) and six general hospitals (Metz-Thionville Hospital, Colmar Hospital, Nord Franche-Comté Hospital, Haguenau Hospital, Verdun Hospital and Mulhouse Hospital) contributed to this study. These nine centers, along with the entire northeast region of France, were heavily impacted by the first wave of the pandemic. As of the end of September 2021, this area reported more than 10,000 deaths due to COVID-19.

From 1 March to 20 April 2020, all adult patients who presented to the ED then directly admitted to the ICU for critical COVID-19 were included in our study. Patients were managed following current guidelines, which, at that time, did not rely on any specific therapeutic strategy. All patients included had at least one nasopharyngeal swab where the RT-PCR was positive for SARS-CoV-2. We only included patients admitted to the ICU following their emergency visit; patients admitted to a conventional medical unit, those admitted to the ICU secondarily during their hospitalization and those managed in ambulatory care were excluded. Additionally, patients who had no positive swab during their hospital stay and those who were subject to limitation of therapeutic effort (including efforts of withdrawing or withholding life-sustaining therapy) were also excluded from the study.

### 2.2. Data Collection

We retrospectively compiled data from patients’ electronic medical records and then standardized them in a report file. The collected data included epidemiological, clinical and biochemical characteristics. Symptom onset date was recorded. Patients’ current treatments and medical history, including cardiovascular disease, diabetes, chronic kidney disease, immune and malignant diseases, were also collected. In this study, severity of disease was defined by admission to the ICU. Obesity was defined by a body mass index superior to 30 kg/m^2^. Overweight was defined by a body mass index between 25 and 30 kg/m^2^. Standard biochemical parameters, such as creatinine, C-reactive protein (CRP), total leukocytes and lymphocytes, were also recorded. We then studied the overall hospital length of stay: firstly, in the ICU, including the need for invasive support care (mechanical ventilation, dialysis, etc.) and the occurrence of complications (pulmonary embolism); then, secondly in post-ICU departments. Severity of illness was assessed using the Simplified Acute Physiology Score II (SAPS II) [10]. Autonomy was measured by the Knaus chronic health status score [11]. We also recorded ICU and in-hospital mortality. Lastly, we studied the impact of age and comorbidities in ICU patients infected with SARS-CoV-2. Therefore, we divided population into three age ranges (<65 years, 65–75 years, >75 years). As for comorbidities, they were grouped into five, according to general systems (cardiovascular, respiratory, hepatic, renal and immune or onco-hematological). All collected data are summarized in the Results Section.

### 2.3. Ethics

This study was approved by the local ethics committee of the University of Strasbourg in France (reference CE: 2020-118), which, in accordance with the French legislation, waived the need for informed consent of patients whose data were entirely retrospectively studied.

### 2.4. Statistical Analysis

The descriptive analysis for categorical variables was executed by providing the frequency of each value. As for continuous variables, the analysis was done by giving median, first and third quartiles. Mann–Whitney U tests were performed to compare the continuous covariates. The Chi-squared test or Fisher’s exact test were performed to compare the categorical covariates in case of expected values below 5 in any of the cells of the contingency tables. In addition, a multivariate logistic model was performed with all the statistically significant and clinically relevant covariates. For survival analysis, survival curves were performed with the use of the Kaplan–Meier method, and the comparison between groups was performed using the log-rank test. Analyses were executed with R 4.0.2 software, as well as with all the software packages required to carry them out.

## 3. Results

### 3.1. Clinical Characteristics of the Study Population

During the study period, which lasted almost two months, 72,941 patients presented to the EDs of all study centers combined. Of these patients, 9296 were diagnosed with SARS-CoV-2 infection using RT-PCR on nasopharyngeal swab and 6346 were hospitalized for COVID-19.

Among all COVID-19 patients, 5570 presented mild disease and 776 presented severe or critical disease and were admitted to the ICU. After excluding patients with missing data and those admitted to the ICU secondarily during their hospitalization, a total of 423 patients were included in our study (Figure 1).

Most patients were male (73.5%, CI 95%: 69.3%–77.7%), median age was 66 years (58–72 years) and almost one-quarter of the study population was overweight (30.3%). In terms of underlying medical conditions, over half of patients (55.6%, CI 95%: 50.8%–60.3%) had high blood pressure, over a quarter of them (27.9%) had a history of diabetes and 16.5% (CI 95%: 12.8%–19.8%) of them presented chronic renal disease. Nearly one-third of the patients included in the study had a history of cardiovascular disease (31.9%, CI 95%: 27.5%–36.6%), nearly one-quarter had a history of respiratory disease (22.5%, CI 95%: 18.5%–26.4%) and only 12.3% (CI 95%: 9.2%–15.4%) of them had a history of onco-hematological disease or immunodeficiency. All characteristics of the study population are summarized in Table 1.

### 3.2. Comparison and Correlation between Survivors and Non-Survivors

Non-survivors were significantly older (69 versus 64 years, *p* < 0.001) than survivors. Although a history of cardio-vascular disease was more frequent in the non-survivor group (*p* = 0.015), other medical history did not differ much in the two groups. Most patients presented normal levels of autonomy as measured by the Knaus score (Knaus score = 6), without any significant difference between the two groups (*p* = 0.251). At ED admission, clinical presentation, laboratory and radiological findings were identical between the two groups, except for the first oxygen saturation measured (*p* = 0.005), creatinine (*p* = 0.001), arterial oxygen tension (*p* = 0.037) and lactatemia (*p* = 0.002). In regard to ventilatory strategies, 377 patients were intubated during their stay (89.1%, IC95%: 86.2%–92.1%), nearly half of whom were intubated in the ED (44.6%). Regarding ICU stay, the severity, as measured by the SAPS II score, was greater in the non-survivor group (47 vs. 40, *p* < 0.001). Non-survivors presented ARDS (*p* = 0.024) more often and were more frequently placed in prone position (*p* = 0.022) or dialyzed (*p* < 0.001) during their ICU stay. As naturally predictable, in the non-survivor group, the length of stay in-hospital and in the ICU shorter as was duration of mechanical ventilation (*p* < 0.001). All comparison results are summarized in Table 1.

On multivariable analysis, age appeared to be a strong predictor of in-hospital mortality. Considering age ranging below 65 years as a reference, the group of age ranging between 65 and 74 years was predictive of mortality (OR = 2.962, CI 95%: 1.231–7.132, *p* = 0.015), whereas the following group of age ranging from 75 to 84 years was not significantly predictive (*p* = 0.177). The group of age ranging above 75 years showed a tendency but was not predictive of in-hospital mortality (OR = 3.084, CI 95%: 0.952–9.992, *p* = 0.06). Conversely, gender (male) as well as almost all the comorbidities when analyzed individually were not predictive of in-hospital mortality, except for immunodeficiency (OR = 4.207, CI 95%: 1.006–17.586, *p* = 0.049). All multivariate analyses are summarized in Table 2.

### 3.3. Overall Survival

Mortality in the ICU was 24.6% (CI 95%: 20.6%–29%) and 26.5% (CI 95%: 22.3%–31%) in-hospital. Survival follow-up was obtained for the entire study population, which allowed us to generate a survival curve, allowing us to visualize mortality over 120 days. Thus, in this cohort of patients admitted in the ICU after presenting to the ED for severe or critical COVID-19, survival at 60 days and 120 days was 65% and 55%, respectively (Figure 2).

### 3.4. The Impact of Age on In-Hospital Mortality

We studied the association of in-hospital mortality and age on one side, then comorbidities on the other. When taking as reference the age group ranging below 65 years, mortality increased regularly and significantly in a nearly linear manner for each age range (Figure 3). An additional survival curve was generated for these three different age ranges, it showed that survival decreased significantly for each group (*p* = 0.004 and *p* = 0.001) when compared to the reference age, ranging below 65 years (Figure 4).

Regarding comorbidities, when taking as reference the absence comorbidities, there seemed to be a lesser tendency for mortality to increase when the number of associated comorbidities increased, and comparison results were not regularly significant (Figure 3).

## 4. Discussion

We focused our study on the most severe patients infected with SARS-CoV-2 infection who were admitted to the ICU during the first wave of the pandemic in France (March–April 2020). In consistence with recent publications thus far, we demonstrated that age was an important predictive factor of in-hospital mortality [5,12,13]. Our results reaffirm the high mortality in elderly COVID-19 patients during this first wave of the pandemic. This allows us to discuss the relevance of access to the ICU and organ support for this elderly population when intensive care capacities and demands hit crisis level.

Over the last 20 years, the median age of patients in the ICU increased and reached over 65 years [14]. These elderly patients were often frail and presented several comorbidities, resulting a decline in their physical health, polymedication and loss of autonomy [15,16]. In comparison to the younger ones, older aged groups reserve capacity is low, leading to an increased risk of mortality when acute medical events occur, such as SARS-CoV-2 infection. Hence, these elderly patients are more likely to develop sepsis, acute heart or respiratory failure [17]. These acute conditions usually present more critically and are more likely to result in secondary organ failure, in fine, increasing mortality rate in older age groups [18].

Intensive care management and its benefit in elderly populations has been long debated [19]. In order to confront the first wave, health care systems worldwide were reorganized due to the pandemic, postponing and canceling all non-essential elective surgical procedures and significantly increasing ICU capacities to cope with the massive load of COVID-19 patients requiring mechanical ventilation. This pressure to increase critical care capacities was inevitably associated with a shortage in human resources (especially nurses) and the creation of temporary and ‘ephemeral’ ICUs on premises not designed for critical hospitalization, which impacted the quality of care provided [8,20]. Since the ICU beds are not infinitely expandable, this pandemic has highlighted the need to establish robust triage and ICU admission criteria. Many protocols and authors attempted to address this delicate issue. Even published national guidelines contained, for the first time, age and frailty scores [21]. These scores did not aim to restrict access of elderly patients to the ICU, but rather to identify those who will benefit most from intensive care, a crucial triage task when ICU admission pressure was on high. Since mortality increased with age and post-ICU follow-up was often complex in older patients, age could be considered as an isolated criterion of non-admission to the ICU [22]. However, age is often associated with different levels of frailty, cognitive decline, sarcopenia and organ failure; thus, it does not seem relevant nor ethical to select patients solely on this criterion. The extensive heterogeneity in practices visible at the beginning of the pandemic should lead us to implement simple, reproducible and reliable ICU admission criteria [23]. In the events of a global health crisis, age, patient autonomy and adequacy of treatment play a role in ICU admission decisions; however, shortage of resources may also play an important part in that decision. Furthermore, ICU admission does not automatically imply the use of invasive techniques; hence, the increase in intermediate care units, which might soon become an essential part of patient management [24]. Finally, as shown by our survival curves and mortality by age range data, the consequences of a severe or critical infection in elderly patients were catastrophic; therefore, it is necessary to elaborate another strategy, notably preventive, for the management of these elderly patients.

Since the start of the COVID-19 pandemic, age was highlighted as one of the major prognostic factors, which was confirmed by our data [12,13]. Indeed, the elderly are more susceptible to infections and develop more severe forms of the disease. This could be explained by physiological aging, particularly that of the immune system (immunosenescence), significant comorbidities and frailty [25]. Thus, as age advances, disruption of both innate and adaptive arms of the immune system is reported. In addition, elderly patients exhibit continuous production of inflammatory mediators and cytokines, also known as ‘inflammaging’ [26]. Immunosenescence is characterized by chronic inflammation (innate/adaptive immune imbalance) and commonly presents in elderly patients. SARS-CoV-2 infection further aggravates the existing inflammatory process and lymphocyte depletion, leading to systemic inflammatory responses [27]. Moreover, immunogenicity of vaccines is often worse in older adults as a result of immunosenescence [28].

In light of our results, having analyzed one of the most severe and critical cohorts during the first wave of the COVID-19 pandemic, and having demonstrated the important impact of age on intra-hospital mortality, it appears essential to protect these elderly fragile populations as much as possible upfront by preventing infection. Despite the emergence of new treatments, alternatives to invasive mechanical ventilation and numerous published studies pointing out age as a risk factor for mortality; we, surprisingly, did not witness a decrease of mortality in elderly populations between the first and second waves of this pandemic [29]. Thus, a privileged or even mandatory access to COVID-19 vaccination should be discussed for these patients, and the current implementation of a vaccine booster with a third dose should particularly target them. These observations provide useful insights in case of the emergence of vaccine-resistant variants or in case of a future health crisis; elderly patients should be targeted in the implementation of all preventive measures and access to vaccination.

## 5. Limitations

Our study presents some limitations such as its retrospective nature and the relatively small sample size. Given the overwhelming workload submerging healthcare systems during the first wave of the pandemic, some of the collected data could not be exhaustively detailed, such as certain parameters (especially during ICU stay) and medical history. This lack of statistical power probably caused insufficient precision when studying different comorbidities as predictive factors of mortality in our cohort. Moreover, data were collected at the very start of the pandemic. At that time, patients were mostly managed following current guidelines, which did not rely on any specific therapeutic strategy since no treatment was validated, vaccines had not yet been marketed and mechanical ventilatory support was almost exclusively invasive due to the fear of aerosolization of viral particles. Since practices and knowledge about this emerging virus have evolved, stays in the ICU are now shorter, invasive mechanical ventilation is less systematic and the effectiveness of vaccines is seeming to curb the pandemic and its critical consequences. Our cohort is undoubtedly of interest since it includes not only the first patients, but also the most critical ones affected by the novel and, at the time, unknown coronavirus. With a hindsight of almost two years now and in light of governmental public health directives (social distancing, quarantine, confinement, vaccination, etc.), this work brings relevant data concerning the management of the elderly and frail in case of another wave of the COVID-19 (vaccine resistant/new variant) or any other future health crisis.

## 6. Conclusions

Age appears to be a relevant predictive factor of in-hospital mortality in cases of severe or critical SARS-CoV-2 infection. We should therefore particularly target and encourage the elderly and more fragile population to comply to preventive health measures (hand washing, masks, social distancing) and prioritize their access to vaccination.

## Figures and Tables

**Figure 1 diagnostics-12-00666-f001:**
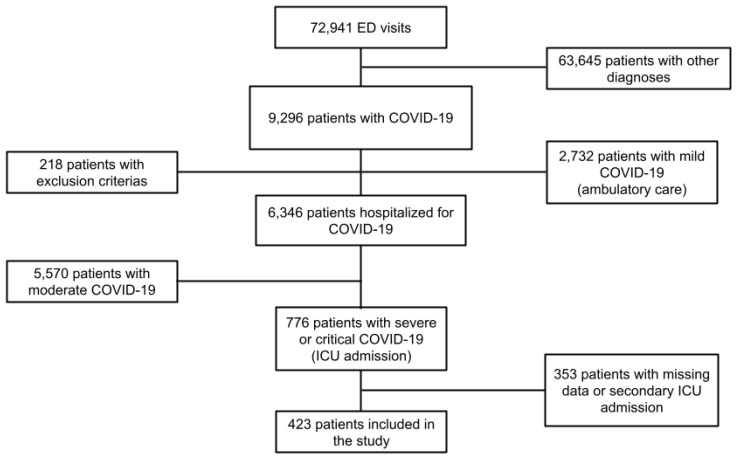
Flowchart of the study. Abbreviations: ED = emergency department, ICU = intensive care unit.

**Figure 2 diagnostics-12-00666-f002:**
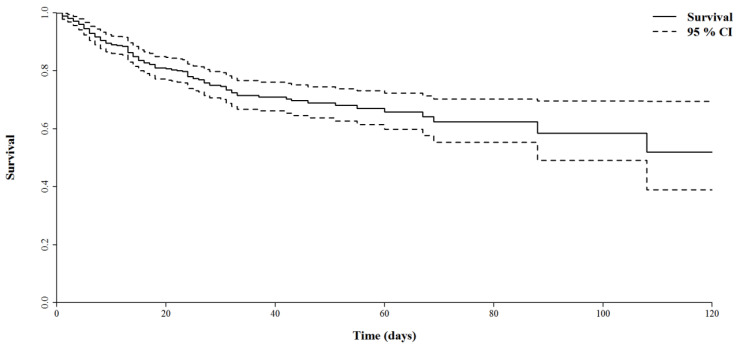
Overall survival of patients admitted to the ICU with a SARS-CoV-2 infection.

**Figure 3 diagnostics-12-00666-f003:**
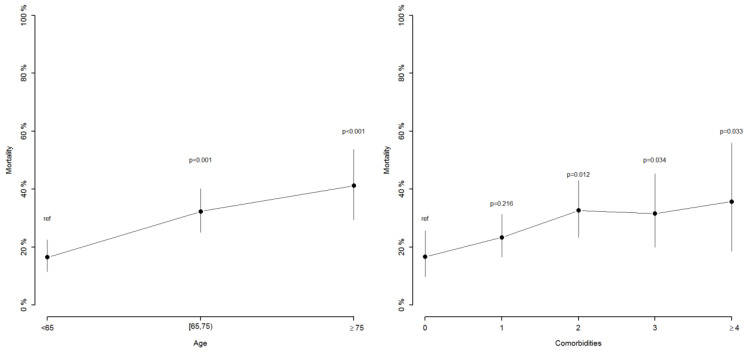
In-hospital mortality according to age and number of comorbidities.

**Figure 4 diagnostics-12-00666-f004:**
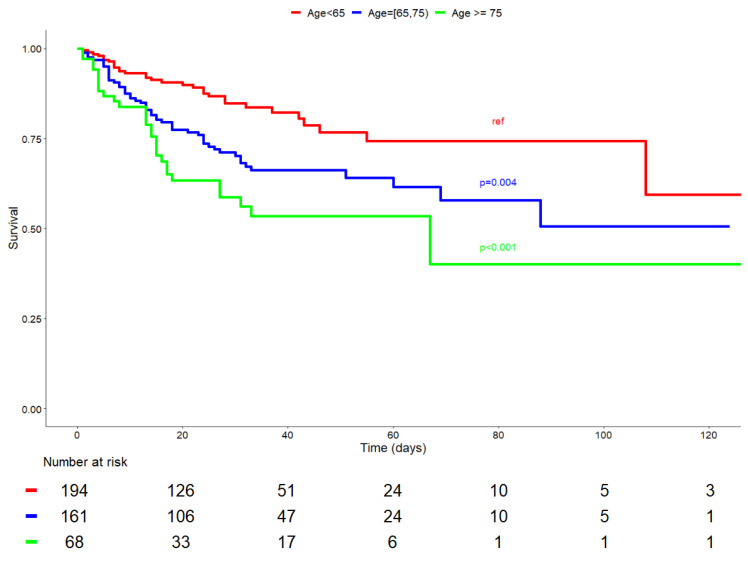
Overall survival of patients admitted to the ICU with a SARS-CoV-2 infection in function of the age.

**Table 1 diagnostics-12-00666-t001:** Clinical characteristics of the study population comparing survivors and non-survivors.

General Characteristics	All Patients *n* = 423	Survivors *n* = 311	Non-Survivors *n* = 112	*p* Value
Age (years)	66 [58–72]	64.0 [56–71]	69 [64–74.3]	<0.001 *
Age <65 years	194 (45.9)	162 (52.1)	32 (28.6)	ref
Age 65–74 years	161 (38.1)	109 (35.1)	52 (46.4)	<0.0001 *
Age >75 years	68 (16.1)	40 (12.9)	28 (25.0)	<0.0001 *
Male *n* (%)	311 (73.5)	222 (71.4)	89 (79.5)	0.096
**Comorbidities *n* (%)**				
Hypertension	235 (55.6)	164 (52.7)	71 (63.4)	0.052
Overweight (BMI > 25 & <30)	125 (30.3)	93 (30.5)	32 (29.6)	0.867
Obesity (BMI > 30)	174 (42.1)	131 (43.0)	43 (39.8)	0.571
Cardiovascular diseases	135 (31.9)	89 (28.6)	46 (41.1)	0.015 *
Diabetes mellitus	118 (27.9)	82 (26.4)	36 (32.1)	0.243
Chronic kidney disease	69 (16.5)	45 (14.6)	24 (22.0)	0.071
Malignancies or ID	52 (12.3)	35 (11.3)	17 (15.2)	0.278
Respiratory diseases	95 (22.5)	69 (22.2)	26 (23.2)	0.823
Total autonomy	386 (91.7)	288 (92.6)	98 (89.1)	0.251
**ED parameters**				
Respiratory rate (/min)	30.0 [24–35]	29 [24–35]	30 [24.5–35.5]	0.218
First oxygen saturation (%)	90 [84–94]	91.0 [85–95.0]	88 [82–92.9]	0.005 *
Oxygen need (L/min)	15 [6–15]	15 [6–15]	15 [9–15]	0.085
Systolic blood pressure (mmHg)	130 [115–142]	130 [115.8–142]	129 [113–145]	0.364
**Laboratory findings**				
Creatinine (μmol/L)	84 [67–105]	80 [66–101.9]	93.0 [72–119]	0.001 *
Lymphocytes (/μL)	780 [580–1110]	790 [600–1128]	725 [500–1063]	0.144
CRP (mg/L)	148.2 [83–223]	147.2 [85–222]	153 [79–223]	0.766
pH	7.46 [7.42–7.49]	7.46 [7.42–7.49]	7.46 [7.41–7.49]	0.456
PaO_2_ (mmHg)	67 [55–80.7]	67.8 [58–82.5]	63.5 [52.6–77.8]	0.037 *
PaCO_2_ (mmHg)	33.9 [30–38]	34 [30.1–38]	33 [28–37]	0.116
Lactate (mmol/L)	1.4 [1.1–2]	1.4 [1–1.9]	1.6 [1.2–2.4]	0.002 *
**Radiological findings *n* (%)**				
Typical CT-scan	223 (53.5)	170 (55.6)	53 (47.8)	0.158
Extension > 50% **ICU stay and outcome**	140 (45.9)	107 (46.1)	33 (45.2)	0.891
SAPS II	42 [32–54]	40 [30.5–51]	47 [39–58]	<0.001 *
Mechanical ventilation (days)	14 [7–24]	15 [8–25]	12 [5–19.5]	0.014 *
ARDS	372 (91.0)	268 (89.0)	104 (96.3)	0.024 *
Prone position	263 (62.5)	183 (59.2)	80 (71.4)	0.022 *
ECMO	16 (3.8)	11 (3.5)	5 (4.5)	0.847
Dialysis	59 (14.0)	31 (10.0)	28 (25.0)	<0.001 *
Pulmonary embolism	50 (11.9)	31 (10.0)	19 (17.1)	0.047 *
ICU LOS (days)	17 [8–30]	19 [10–31]	13 [6–24]	<0.001 *
In-hospital LOS (days)	26 [13–43]	30.0 [19–48]	13.5 [6–24.3]	<0.001 *

Data are all expressed in median [Q1–Q3] or *n*/N (%) where N is the total number of patients with available data. * *p* < 0.05. Abbreviations: BMI = body mass index, ED = emergency department, ID = immunodeficiency, ICU = intensive care unit, CRP = C-reactive protein, SAPS II = Simplified Acute Physiology Score II, ARDS = acute respiratory distress syndrome, ECMO = extracorporeal membrane oxygenation, LOS = length of stay.

**Table 2 diagnostics-12-00666-t002:** Multivariable analysis of factors associated with in-hospital mortality.

General Characteristics	Odds Ratio	95%CI	*p* Value
Age < 65 years	1	ref	
Age 65–74 years	2.962	[1.231–7.132]	0.015 *
Age ≥ 75 years	3.084	[0.952–9.992]	0.060
Gender (male)	2.753	[0.936–8.100]	0.066
SAPS II	1.027	[1.000–1.055]	0.052
**Comorbidities**			
Knaus score < 6	0.739	[0.160–3.400]	0.697
Hypertension	0.880	[0.355–2.181]	0.782
Obesity (BMI > 30)	1.045	[0.411–2.657]	0.926
Overweight (BMI > 25 and <30)	1.273	[0.494–3.284]	0.617
Cardiovascular diseases	1. 248	[0.503–3.100]	0.633
Diabetes mellitus	0.944	[0.367–2.427]	0.905
Respiratory diseases	0.990	[0.328–2.991]	0.986
Asthma	0.739	[0.249–2.193]	0.586
COPD	0.586	[0.216–1.589]	0.294
Liver diseases	0.610	[0.070–5.319]	0.655
Chronic kidney disease	0.853	[0.285–2.551]	0.776
Malignancies	0.842	[0.229–3.098]	0.796
Immunodeficiency	4.207	[1.006–17.586]	0.049 *

Abbreviations: SAPS II = Simplified Acute Physiology Score II, BMI = body mass index, COPD = chronic obstructive pulmonary disease, ID = immunodeficiency. * *p* < 0.05.

## Data Availability

All data analyzed as part of the study are included. The data presented in this study are available on request from the corresponding author.

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
