# Peer review of "The Impact of Age on In-Hospital Mortality in Critically Ill COVID-19 Patients: A Retrospective and Multicenter Study"

_diagnostics, 2022, doi:10.3390/diagnostics12030666_

Round 1
Reviewer 1 Report
Thank you for giving me the opportunity to review this manuscript. In this study, the authors investigated the impact of comorbidity and age on in-hospital mortality of patients with COVID-19. The research question is interesting but there are too many methodological issues that probably led to counterintuitive and paradoxical study results . Additionally, we know that age was considered an important prognostic factor in COVID-19; I don't find any novelty in this; for this reason I suggest the authors to perform the study from the beginning, with the suggestions below:
1) Methodology:
-Categorization of age should be somewhat balanced to avoid introduction of bias due to small number of patients belonging to some groups compared to other ones. In this analysis, you found that patients aged 75-84 years died less than those aged 65-74 years, but you concluded saying that age is a strong prognostic factor, that is not properly correct; why patients aged 75-84 died less than those 65-74 years? I think that WHO categorization should not fit this study population. Indeed, confindence intervals of ORs of the last age group are too wide and add uncertainty to such findings. How many patients are 85 or older? At the light of this, age could not be defined a strong prognostic risk factor in the present study. You could overcome this issue by 1) studying the relationship between age and mortality to assess for non-linearity (the relationship between continous age and mortality is really non-linear? Or it seems to be non-linear because of inadequate categorization?). In case of non-linearity, you could also use spline regression to assess for non-linearity and investigate the existence of a polynomial association. 2) Repeat the analysis, categorizing according to a) tertiles or quartiles of age; b) <65, 65-74, 75 or more.
-Number of patients belonging to age categories should be added to Table 1.
-In statistical analysis and methods, you missed to describe how univariate and multivariate logistic regression models were performed after Kaplan Meier curve presentation. Which variables did you introduce in the multivariate models? To avoid overcorrection of study model, I suggest to introduce more important diseases and laboratory values and not all diseases. Additionally, do you have a measure of cumulative morbidity (CIRS-G, CCI, etc.?).
-Design new survival curves according to new categorizations (<65, 65-74, 75 or more years; or tertiles/quartiles of age).
-Improve description of comorbidities in the data collection section. What stands for 'cardiovascular diseases' and 'respiratory diseases'? 'Chronic kidney diseases' stands for Chronic Kidney Disease (CKD)?
-Comorbidities have previously shown to have a significant impact on COVID-19 survival. However, single diseases showed no or little impact on study outcome. You could add a measure of number of comorbidities presented by each patients (< 4 and 4 or more) to investigate comorbidities by a quantitative aspect, since multiple diseases may favor multiorgan failure development as a cause of death in COVID-19 patients.
-Comorbidities are directly related with age; despite their potential weight was not so evident in the entire population, I would try to investigate the impact of single comorbidities in each age group. At this regard, I would add an additional table stratified by new age group (table 2, with each age group in columns, and number of events and covariates in rows (comorbidities, number of comordities, etc.).
Author Response
Reviewer 1
Thank you for giving me the opportunity to review this manuscript. In this study, the authors investigated the impact of comorbidity and age on in-hospital mortality of patients with COVID-19. The research question is interesting but there are too many methodological issues that probably led to counterintuitive and paradoxical study results. Additionally, we know that age was considered an important prognostic factor in COVID-19; I don't find any novelty in this; for this reason I suggest the authors to perform the study from the beginning, with the suggestions below:
We thank Reviewer 1 for their feedback. We, indeed, aimed to study the impact of age and, to a lesser extent, comorbidities on in-hospital mortality, in a cohort of critical COVID-19 patients during the first wave of the pandemic. We have followed with our point-by-point response to the reviews.
1) Methodology:
- Categorization of age should be somewhat balanced to avoid introduction of bias due to small number of patients belonging to some groups compared to other ones. In this analysis, you found that patients aged 75-84 years died less than those aged 65-74 years, but you concluded saying that age is a strong prognostic factor, that is not properly correct; why patients aged 75-84 died less than those 65-74 years? I think that WHO categorization should not fit this study population. Indeed, confidence intervals of ORs of the last age group are too wide and add uncertainty to such findings. How many patients are 85 or older? At the light of this, age could not be defined a strong prognostic risk factor in the present study. You could overcome this issue by 1) studying the relationship between age and mortality to assess for non-linearity (the relationship between continous age and mortality is really non-linear? Or it seems to be non-linear because of inadequate categorization?). In case of non-linearity, you could also use spline regression to assess for non-linearity and investigate the existence of a polynomial association. 2) Repeat the analysis, categorizing according to a) tertiles or quartiles of age; b) <65, 65-74, 75 or more.
The relation between age and mortality is not linear. Age, ranging from 20 to 60 years old, had no effect on mortality, however, above 60 years old, the relation between age and mortality becomes approximately linear. This is the same multivariate model where age is linear (threshold 60 years old).
|
|
N |
Odds Ratio |
95%CI |
p value |
|
Age ≤ 60 years |
136 |
0.982 |
[0.922-1.045] |
0.559 |
|
Age > 60 years |
287 |
1.111 |
[1.043-1.183] |
0.001* |
For patients aged ≤ 60 years old, the increase of one year in terms of age resulted in a non-significantly increased risk of death, by 0.982 [0.922-1.045].
As requested, we changed and repeated the analysis with patients <65 years, 65-74 years and >75 years. Therefore, Table 2 changes, as does the survival curve.
For patients >60 years old, the increase of one year in terms of age resulted in a significantly increased risk of death, by 1.111 [1.043-1.183].
- Number of patients belonging to age categories should be added to Table 1.
As requested, we have added the number of patients belonging to each of the three age categories to Table 1 and our manuscript.
- In statistical analysis and methods, you missed to describe how univariate and multivariate logistic regression models were performed after Kaplan Meier curve presentation. Which variables did you introduce in the multivariate models? To avoid overcorrection of study model, I suggest to introduce more important diseases and laboratory values and not all diseases.
The variables introduced in the multivariate model were: age, Knaus Chronic health status score below 6 (indicating altered levels of autonomy), SAPS II score, main comorbidities (hypertension, diabetes, respiratory diseases, cardiovascular diseases, chronic kidney disease, malignancies or immunodeficiency, obesity...), along with clinical parameters (such as oxygen saturation) and management (endotracheal intubation in the ED), laboratory findings (pH<7.35, CRP>100mg/l, creatinine level at admission) radiological findings (typical lesions on chest CT scan) and complications (pulmonary embolism, ECMO, TVP).
These variables were introduced into the model either because of their significance in univariate analysis or because of their clinical relevance.
- Additionally, do you have a measure of cumulative morbidity (CIRS-G, CCI, etc.?).
We understand the clinical relevance of these type of scales, such as the clinical frailty scale (CFS), an important predictive factor for death in intensive care. However, unfortunately, at the time of the first wave of the outbreak, we were no position to obtain and calculate a cumulative morbidity score.
- Design new survival curves according to new categorizations (<65, 65-74, 75 or more years; or tertiles/quartiles of age).
We designed new survival curves accordingly.
- Improve description of comorbidities in the data collection section. What stands for 'cardiovascular diseases' and 'respiratory diseases'? 'Chronic kidney diseases' stands for Chronic Kidney Disease (CKD)?
As previously said, at the time of the first wave of the outbreak, we, unfortunately, were no position to obtain detailed medical history for each patient. We mentioned this in the limitations of our manuscript, stressing on the fact that the conditions in which our multicentric data were collected, at the peak of the first wave of COVID-19 in the Great-East of France were not ideal for exhaustive and detailed data collection. This was also the reason why we did not elaborate on comorbidities. However, in multivariate analysis, immunodeficiency appeared to be an important risk factor for mortality (which is now proven, 2 years after the outbreak). Our other results regarding certain comorbidities were probably affected by the lack of proper sample size (obesity, males, …).
Thus, we combined in cardiovascular diseases all cardiovascular diseases (rhythmic, acute heart failure, ischemic), similarly for respiratory diseases (asthma, COPD, all causes together) and for CKD according to creatinine clearance.
- Comorbidities have previously shown to have a significant impact on COVID-19 survival. However, single diseases showed no or little impact on study outcome. You could add a measure of number of comorbidities presented by each patients (< 4 and 4 or more) to investigate comorbidities by a quantitative aspect, since multiple diseases may favor multiorgan failure development as a cause of death in COVID-19 patients.
- Comorbidities are directly related with age; despite their potential weight was not so evident in the entire population, I would try to investigate the impact of single comorbidities in each age group. At this regard, I would add an additional table stratified by new age group (table 2, with each age group in columns, and number of events and covariates in rows (comorbidities, number of comordities, etc.).
We did not obtain detailed and exhaustive medical history for each patient, for the reasons mentioned above. The comments made by Reviewer 1 are, indeed, relevant, particularly regarding the link between age and the increase in comorbidities. However, we wanted to focus our study on the impact of age on this ‘naïve’ cohort of critical COVID-19 during the first wave of the outbreak. Our lack of power in terms of sample size for certain age groups and for comorbidities probably impacted the non-significance of our results.
Reviewer 2 Report
[Table 2] By looking at Table 2, I am not convinced that five categories of age can be a meaningful predictive factor of in-hospital mortality in critically ill COVID-19 patients. Perhaps the result is due to noise or bias due to not randomizing the individuals. I suggest some kind of stratification to find more understandable results.
[Multiple lines] I suggest you clearly state what you mean by 'the first wave' and 'the second wave.' For example, when are each wave and how you defined them.
Author Response
Reviewer 2
[Table 2] By looking at Table 2, I am not convinced that five categories of age can be a meaningful predictive factor of in-hospital mortality in critically ill COVID-19 patients. Perhaps the result is due to noise or bias due to not randomizing the individuals. I suggest some kind of stratification to find more understandable results.
We thank Reviewer 2 for their feedback. This first comment is similar to Reviewer 1’s. Hence, following what was suggested, and in order to improve our manuscript, we changed the title to become less assertive about the impact of age. Additionally, the age categories were changed (Table 2). It appears that our results were affected by the lack of power (small sample size for certain age groups, notably the above 75 years old) which made the aOR non significative for age ranging between 75 and 84 years, whereas in both other age classes, aOR was significative. We also added the numbers for the survival curve so that the reader can better understand the results. Additionally, we wanted to stress that this cohort of critical COVID-19 patients admitted to the ICU, represented a ‘real life’ analysis of a ‘naïve’ cohort of critical COVID-19 during the outbreak’s first wave (March-April 2020), a period where no treatment nor vaccine was available. As of the 2 past years, we now have proper data allowing a better understanding of the risk factors of SARS-CoV-2 infection. Age and comorbidities (such as immunodeficiency) are now widely demonstrated as risk factors of mortality in COVID-19. Despite this, we believe that the message we presented straightforward, namely, promoting the access to vaccination and discussing the relevance of ICU admission in the elderly patients in case of high ICU admission demand, in a context of health crisis and shortage of critical care resources.
I suggest you clearly state what you mean by 'the first wave' and 'the second wave.' For example, when are each wave and how you defined them.
The precise timeline of each wave of the COVID-19 pandemic varied greatly depending on the country and region studied. We clarified our manuscript by specifying the timelines, especially that of the study (March-April 2020), a period corresponding to the first wave of the outbreak in our region, the Great-East of France.
Round 2
Reviewer 1 Report
The authors addressed all my concerns and substantially improved description of methods and presentation of study results. I think that now the manuscript is ready to be published, after minor editorial checks. In summary, age range 65-74 years was a major risk factor for mortality compared to being < 65 years; individuals aged 75 years or more showed a trend of increased risk without reaching statistical significance. Net of any potential confounders included in fully adjusted models, age represents a major risk factor for mortality in critically ill patients with COVID-19. This corroborates the importance of early treatment and accurate monitoring of patients in this age range.
This manuscript is a resubmission of an earlier submission. The following is a list of the peer review reports and author responses from that submission.